# Dicing the Disease with Dicer: The Implications of Dicer Ribonuclease in Human Pathologies

**DOI:** 10.3390/ijms21197223

**Published:** 2020-09-30

**Authors:** Eleni I. Theotoki, Vasiliki I. Pantazopoulou, Stella Georgiou, Panos Kakoulidis, Vicky Filippa, Dimitrios J. Stravopodis, Ema Anastasiadou

**Affiliations:** 1Center of Basic Research, Biomedical Research Foundation of the Academy of Athens (BRFAA), 11527 Athens, Greece; elthk@biol.uoa.gr (E.I.T.); vaspantazo@bioacademy.gr (V.I.P.); sgeorgiou@bioacademy.gr (S.G.); pk88gr@hotmail.com (P.K.); vickyrougefilippa@gmail.com (V.F.); 2Section of Cell Biology and Biophysics, Department of Biology, School of Science, National and Kapodistrian University of Athens (NKUA), 15701 Athens, Greece; dstravop@biol.uoa.gr

**Keywords:** Dicer ribonuclease, post-transcriptional regulation, RNA binding proteins, microRNAs, RISC, Dicer1 syndrome, human diseases, cancer, neurological disorders, autoimmune diseases, viral infection, infertility, cardiovascular diseases

## Abstract

Gene expression dictates fundamental cellular processes and its de-regulation leads to pathological conditions. A key contributor to the fine-tuning of gene expression is Dicer, an RNA-binding protein (RBPs) that forms complexes and affects transcription by acting at the post-transcriptional level via the targeting of mRNAs by Dicer-produced small non-coding RNAs. This review aims to present the contribution of Dicer protein in a wide spectrum of human pathological conditions, including cancer, neurological, autoimmune, reproductive and cardiovascular diseases, as well as viral infections. Germline mutations of *Dicer* have been linked to Dicer1 syndrome, a rare genetic disorder that predisposes to the development of both benign and malignant tumors, but the exact correlation of Dicer protein expression within the different cancer types is unclear, and there are contradictions in the data. Downregulation of Dicer is related to Geographic atrophy (GA), a severe eye-disease that is a leading cause of blindness in industrialized countries, as well as to psychiatric and neurological diseases such as depression and Parkinson’s disease, respectively. Both loss and upregulation of Dicer protein expression is implicated in severe autoimmune disorders, including psoriasis, ankylosing spondylitis, rheumatoid arthritis, multiple sclerosis and autoimmune thyroid diseases. Loss of Dicer contributes to cardiovascular diseases and causes defective germ cell differentiation and reproductive system abnormalities in both sexes. Dicer can also act as a strong antiviral with a crucial role in RNA-based antiviral immunity. In conclusion, Dicer is an essential enzyme for the maintenance of physiology due to its pivotal role in several cellular processes, and its loss or aberrant expression contributes to the development of severe human diseases. Further exploitation is required for the development of novel, more effective Dicer-based diagnostic and therapeutic strategies, with the goal of new clinical benefits and better quality of life for patients.

## 1. Introduction

The Dicer enzyme, a well-conserved protein among eukaryotic organisms [1,2], is a large protein (~200 kDa), initially identified in *Drosophila melanogaster* [3]. Most of higher metazoa, including humans, have a unique *Dicer* gene in their genome [1,4] whose product is an endonuclease (a member of the ribonuclease III (RNase III) family) [5,6,7,8]. Mammalian Dicer structure, although difficult to crystallize [9], has been inferred via cryo-electron microscopy and biochemical and crystallographic studies on individual domains of the protein, which resembles the shape of the letter L, with a head, a body and a base [1,10,11]. Dicer main domains, ordered from the N- to the C-terminus, are helicase domain (including DExD/H, TRBP-BD and HELICc), DUF283 domain, PAZ (Piwi/Argonaute/Zwille) domain, RNase IIIa and RNase IIIb domains and dsRNA-binding domain (RBD) [9,10,12,13]. Through these domains, Dicer is involved in canonical biogenesis of most small regulatory RNAs, including microRNAs (miRNAs) (Figure 1) and small interfering RNAs (siRNAs). Specifically, Dicer cleaves twice the precursor miRNA (pre-miRNA) hairpins at the stem-loop boundary, generating mature miRNA [14,15], a small non-coding RNA (nc-RNA) of ~22 nucleotides in length that is characterized by a 2-nucleotide overhang at the 3′-end [16]. In mammals, TAR-binding protein (TRBP) and PKR activator (PACT) compose, together with Dicer, the RISC loading complex [17,18,19]. Argonaute proteins (AGOs), and especially AGO2, also constitute RISC loading complex [20] by binding to the C-terminal region of human Dicer. AGO2, together with the mature miRNA, composes the miRNA-induced silencing complex (miRISC) [13,21].

Dicer is also involved in generating mature miRNAs from other RNA species, such as non-coding small nucleolar RNAs (snoRNAs) [22] and transfer RNA (tRNA)-related fragments (tRFs) [23]. Interestingly, while Dicer is localized and functions in the cytoplasm, there is evidence for additional tasks into the nucleus [24,25] and into the nucleolus, with a potential role as a tumor suppressor [26]. Nuclear Dicer has been associated with transcriptional silencing, RNA post-transcriptional processing, DNA damage response and dsRNA removal [1,27,28,29,30]. Furthermore, other studies have indicated the involvement of Dicer in autophagy and autophagosome formation [31,32], stabilization of passive-site RNAs [33], antiviral defense [34,35,36] and apoptosis [37,38].

Evidence for serious developmental abnormalities and baneful human diseases such as cardiovascular diseases and cancer [10], caused by loss or aberrant expression of Dicer protein, has emerged. Deep understanding of the localization patterns, expression alterations and mutations of Dicer in disease states, as well as its post-translational modifications, will allow precise molecular targets to be identified for the design of novel, more effective therapeutic approaches. In this review, the leading Dicer-associated human disorders are presented, demonstrating both the great impact of the depletion or the overexpression of Dicer in cells and the potential of this molecule to be exploited as a biomarker or/and therapeutic target for several human diseases.

## 2. Dicer in Human Diseases

### 2.1. Dicer1 Syndrome

Dicer1 syndrome—also known as pleuropulmonary blastoma familial susceptibility syndrome—is a rare genetic disorder that is inherited in an autosomal dominant manner [39] and predisposes the development of both benign and malignant tumors [37,40]. Pathogenic germline mutations of the *Dicer* (also known as *Dicer1*) gene have been linked to Dicer1 syndrome (Figure 2), whereas mosaic mutations of this gene have also been associated with this condition [37,41]. Mutations in one gene allele lead to an increased risk of tumor development, although, in many cases, this is not sufficient to cause a malignant phenotype. The majority of these mutations are located within regions that encode the main protein’s domains (Figure 3A,B) and usually result in amino acid alterations and loss of function [37,39].

To date, Dicer1 syndrome has been associated with pleuropulmonary blastomas, cystic nephroma, rhabdomyosarcoma, multinodular goiter, thyroid cancer, Sertoli–Leydig cell tumor, ovarian sex cord-stromal tumor, ciliary body medulloepithelioma, pituitary blastoma, pineoblastoma and other neoplasms [37,42,43,44,45]. Furthermore, in rare cases, neuroblastoma, Wilms’ tumor and other more common childhood cancers have also been reported [39]. Although Dicer1 syndrome is an autosomal-dominant disease inherited in a haploinsufficient manner, recent studies have indicated the necessity of a somatic mutation in the second allele of the *Dicer* gene, in addition to a preexisting germline mutation in one allele (two-hit hypothesis) [37,46]. These “second-hit” mutations have been found in the regions encoding the RNase III domains, which are genetic hotspots for somatic mutations within the *Dicer* gene [47,48].

Metformin, an oral hypoglycemic agent, was found to reduce tumor growth with a simultaneous enhancement of *Dicer* gene expression in mice. The increase of Dicer levels both in human patients and in mouse models occurred by altering the subcellular localization of AUF1, a *Dicer* mRNA binding protein, increasing Dicer mRNA stability and allowing the latter to accumulate [49,50]. Metformin therapeutic approaches might be more beneficial for patients with a single functional *Dicer* allele, but not for patients with biallelic *Dicer* mutations [37]. However, to improve success regarding the treatment of Dicer1 syndrome, a functional combination of basic, translational and clinical research will be required.

### 2.2. Dicer and Other Cancer Types

*Dicer* gene expression seems to be significantly associated with several cancer types, such as lung, ovarian, colorectal and prostate cancer. Although the implication of Dicer in tumorigenesis has been indicated, the exact correlation of Dicer protein expression with the different cancer types is unclear, as the existing reports are contradictory and do not necessarily correlate with mRNA and protein expression of Dicer as presented in the Cancer Genome Atlas (TCGA) (Figure 4A–C). The observed differences between malignancies are, most likely, due to the deregulation of tissue-specific miRNA expression, probably according to their susceptibility to Dicer, resulting in defective cell/tissue growth and function. Reduced Dicer expression has been linked to breast and lung cancer, while lower levels of this protein have been associated with poor prognosis and higher tumor stage in these patients [10,51,52,53,54].

In some cases of non-small cell lung carcinomas (NSCLC), Dicer expression level has significantly reduced during stage I, but risen in later stages [53]. Complicating matters further, Dicer is overexpressed in prostate and colorectal cancer and acute lymphoblastic leukemia (ALL), leading to a shorter survival rate [55,56,57]. In ovarian cancer cases, both reduced levels and overexpression of Dicer have been reported [58,59]. In ovarian tumors, loss of Dicer has been demonstrated to promote cell proliferation and migration with a simultaneous decreasing sensitivity to cis-platin [60]. Conflicting data also exist regarding thyroid cancer, as Dicer has been found both upregulated [61] and downregulated in independent patient cohorts [62]. The alterations of Dicer levels and the consequent aberrant expression of specific miRNAs, with a prominent role in these tissues, may be driving this phenotype. Particularly, miR-21 [63] and miR-4661-5p [64] have been found to act as oncogenic miRNAs (onco-miRs) in colorectal cancer and hepatocellular carcinoma, respectively, whereas miR-124-5p [65] and miR-26b-5p [66] seem to protect against to colorectal and thyroid cancer, respectively.

Moreover, Dicer is involved in cell responses due to double-strand breaks (DSBs) [28] for the maintenance of genome integrity and survival, revealing another mechanistic role of this molecule in cancer. Given this, Dicer probably processes RNAs formed by the transcription of broken DNA ends [67]. It has been demonstrated in *Arabidopsis* and in human cells that small RNAs are produced from sequences flanking DSB sites [68] named diRNAs. The latter function as guide molecules that mediate chromatin modification or recruit protein complexes facilitating DSB repair [68].

Based on this evidence, the multifactorial role of Dicer in cancer is becoming apparent, although there is no clear correlation between Dicer expression and cancer type or/and progression. A thorough investigation of the implications of Dicer on the different malignancies is necessary to provide an in-depth knowledge regarding its action in tumorigenesis. Through this process, the ability of Dicer to act as either a tumor suppressor or an oncogene, according to the cancer type, will be clarified, and its potential application as a diagnostic/prognostic tool or a therapeutic target will most likely emerge.

### 2.3. Dicer in Geographic Atrophy

Geographic atrophy (GA) is an advanced form of age-related macular degeneration, a severe eye-disease that is a leading cause of blindness in industrialized countries [69,70,71,72]. In GA patients, Dicer levels have been reduced in the macular retinal pigment epithelium (RPE)—an eye-specific tissue that is affected in GA—but not in the neural retina [69,71]. The downregulation of Dicer is not a generic damage response due to the chemical stress caused by dying cells, but is GA specific, as the protein levels of Dicer have not been reduced in patients with other RPE disorders. Experiments in mice have shown that Dicer deletion results in RPE cell degeneration. Moreover, Dicer downregulation in mouse and human RPE cells increases cell death, indicating the involvement of Dicer deregulation in the pathogenesis of the disease (Kaneko, Dridi et al. 2011).

Given that Dicer plays a key role in the biogenesis of miRNAs, and that these molecules are abundant and important in mammalian cells, miRNA dysregulation due to Dicer depletion would be expected in the occurrence of GA. Surprisingly, this is not the case, as the deletion of other miRNA-processing enzymes does not lead to RPE degeneration. Existence of replicated *Alu* RNA sequences—non-coding RNAs expressed by highly abundant *Alu* retrotransposon [73,74], specifically in the RPE of examined patients—has been observed as a result of the absence of Dicer enzyme. Normally, Dicer processes *Alu* RNAs into shorter, non-toxic molecules, whereas the abundance of *Alu* RNAs induces cell death in RPE cells and RPE degeneration [71], possibly via the activation of the NLRP3 inflammasome [73]. These data support a model according to which GA is likely to be mechanistically associated with Dicer deregulation and *Alu* RNAs homeostasis, providing a scaffold for the development of new therapeutic strategies.

### 2.4. Dicer in Psychiatric and Neurological Diseases

#### 2.4.1. Chronic Stress and Depression

Chronic stress, a condition commonly associated with depression, can cause changes in the endocrine system, as well as alterations in the expression of genes related to stress response [75,76]. Chronic stress is both a psychological and neurobiological phenomenon, and has been shown to alter the hypothalamo-pituitary-adrenal (HPA) system [75,77]. It may also affect hippocampus and prefrontal cortex volume, as well as dendritic growth in the amygdala, leading to impairment of memory and emotional control, and augmentation of anxiety and aggression, respectively [78]. During stress and depression, many biological processes and components undergo changes, with the serotonergic system, the Wnt signaling pathway and β-catenin expression being typical examples [75]. β-catenin, a fundamental component of the canonical Wnt signaling pathway, is a ubiquitously expressed protein in the mammalian brain that is involved in both intercellular adhesion and gene transcription [79,80]. Its dysfunction has been implicated in several neuropsychiatric disorders, such as depression. Specifically, a downregulation of the transcription output of β-catenin has been demonstrated in patients with depression, whereas the overexpression of this molecule in mouse depression models has resulted in the prevention of the social avoidance phenotype [81]. Given that β-catenin appears to bind and regulate *Dicer* gene expression, its behavioral effects have also been investigated experimentally (Figure 5). Dicer knockdown has led to a social avoidance phenotype, similar to that caused by blocking β-catenin signaling. In addition, β-catenin overexpression has blocked the development of this phenotype in mice expressing normal Dicer levels, but not in Dicer downregulated mice, indicating that β-catenin acts—at least in a part—via the Dicer enzyme [81].

*Dicer* mRNA levels have also been found decreased in post-traumatic stress disorder (PTSD) with comorbid depression, a distinguished category of chronic stress [82]. Towards the same direction, a follow-up fMRI study uncovered that lower *Dicer* blood mRNA levels were significantly associated with increased amygdala activation to fearful stimuli, a neural response correlated with PTSD [83]. Stress-related regulation of *Dicer* and miRNAs in the blood and the brain of these patients may occur simultaneously, as the findings of stress-related *Dicer* and miRNAs are consistent in blood and into the nucleus accumbens [83]. This brain region of chronic stress mice is responsible for motivation, pleasure, addiction and learning [81]. Additionally, Dicer expression quantitative trait locus (eQTL), rs10144436, located in its 3′ untranslated region (3′ UTR), is significantly associated with PTSD and depression [84], inferring that Dicer plays a role in the mechanism or/and manifestation of the abovementioned condition [83]. As the differential expression of many miRNAs has been associated with organisms’ adaptations under stress conditions and depression [85,86,87], miRNAs probably mediate the behavioral effects of Dicer. The lack of the latter and the subsequent deregulation of miRNA expression could cause severe alterations in neural processes, an increased sensitivity to stress and the development of depression, as a final result. In any case, the Dicer function seems to be important for the prevention of depression, as it seems to play a key role in the adaptive responses to cell stress.

#### 2.4.2. Parkinson’s Disease

In the human brain, neuronal networks formed by dopaminergic (DA) neurons are responsible for the regulation of emotions, complex behaviors and voluntary motion [88]. A progressive damage to DA neurons in the midbrain substantia nigra (SN) can affect motor symptoms in Parkinson’s disease (PD), an age-related neurodegenerative disorder that affects approximately 1% of the population older than 65 years old [89].

Experimental data and scientific evidence support a strong implication of Dicer affecting PD (Figure 5). During aging, Dicer has been found downregulated in different tissues, including brain [90,91], a reduction observed in DA neurons from PD patients [92]. In vivo studies have shown that Dicer knockout in DA neurons leads to increased cell death and degeneration of their axonal projections to the striatum, following a pattern of cell death similar to clinical PD manifestation. Noticeably, the DA neuron-specific ablation of Dicer in mice has caused balance and motor coordination abnormalities that worsened with time, leading to a progressive development of a PD-like phenotype [93,94]. Furthermore, loss of Dicer results in a complex of PD symptoms characterized by profound involuntary resting tremor, postural and gait impairments and rigidity. However, increasing the activity of Dicer in these animals via pharmacological intervention with enoxacin—a drug known to stimulate Dicer activity—was neuroprotective, indicating the crucial role of this protein in the development and progression of the disease [93]. Loss of Dicer in the brain confers higher levels of pro-inflammatory factors and hyper-microglial inflammatory responses [95]. Neuroinflammation seems to contribute to the development of PD [94,96,97], thus this may be a possible mechanism through which Dicer is involved in disease progression. Hence, Dicer could be a potentially promising target for the treatment or/and the alleviation of symptoms of PD.

#### 2.4.3. Triplet Repeat Expansion Diseases

Triplet repeat expansion diseases (TREDs) are a class of several genetic disorders, including Huntington’s disease (HD), myotonic dystrophy type 1 (DM-1) and fragile X syndrome, caused by triplet repeat expansions in specific genes [98]. They are correlated to elongation of CNG DNA and RNA repeats, whose length has varied between patients and healthy individuals, encoding for a stretch of glutamines (polyQ) that are responsible for the toxic effects in the cells [99]. These triplet repeats form long hairpin structures [100,101], which are recognized and cleaved by Dicer enzyme, resulting in the generation of short CNG RNAs of approximately 21 nucleotides in length (siCNGs) [99,102,103] (Figure 5). The relation of siCNGs and HD has been previously reported, due to a CAG expansion within the first exon of the Huntingtin (HTT) gene. These Dicer-dependent siCAGs from HTT RNA seem to have neuro-toxic effects [103]. siCNG levels have been increased in brain samples isolated from patients with HD compared to healthy individuals [103]. Moreover, the toxic activity of Dicer-dependent CNG biogenesis has been documented in fibroblasts of patients with DM-1 (siCUG) and HD (siCAG) [99]. Also of note, the experimentally measured levels of the TRED-related transcripts in cells with normal expression of Dicer were lower compared to Dicer-deficient cells [99]. Accordingly, the Dicer-induced siCNGs can silence the expression of transcripts containing long complementary repeats via the RNA interference pathway [99,104,105]. This indicates the existence of small molecules in TREDs that can direct the repression of specific targets, leading to potentially pathogenic gene expression unbalances. Given that the presence of these siRNAs could cause neuro-toxic effects [103], the discovery of the Dicer-regulated siCNGs mechanism highlights the potential value of these molecules as promising therapeutic targets for triplet repeat expansion diseases.

### 2.5. Dicer in Autoimmune Disorders

#### 2.5.1. Psoriasis

Psoriasis, a papulosquamous skin disease, is a common immune-mediated disorder [106]. Specifically, psoriasis is a chronic skin inflammatory disease characterized by a vicious circle of chronic inflammation caused by the interaction between keratinocytes and immune cells [107,108]. The pathogenesis of psoriasis involves both innate and acquired immunity, as well as genomic background, keratinocytes and environmental factors [109,110]. Transcripts of approximately 1340 genes were found to be deregulated, with potential implications for this disorder [106,107,111]. In a recent study, the expression level of Dicer was also deregulated in skin biopsies, demonstrating higher levels of transcription in psoriasis skin tissue than in healthy individuals [106] (Figure 6). Dicer appears to have a distinct role in psoriasis, and the aberrant expression of this molecule could be related to disease progression. This may be due to the subsequent deregulation of gene expression and small RNAs, as several miRNAs have been found to be deregulated in psoriasis skin lesions [112]. miRNAs are involved in skin development, epidermal differentiation and hair follicle development [113,114], whereas individual miRNAs control the levels of inflammatory proteins in the skin of patients with psoriasis [106]. Among others, miR-203 is a typical example of a tissue-specific miRNA, which has a skin-restricted expression profile and acts as a tumor suppressor regulating the differentiation and proliferation of keratinocytes. Thus, both miRNAs and RNAi machinery proteins, including Dicer, seem to be issues of major importance for the occurrence and progression of psoriasis and, consequently, for the diagnosis, prognosis and treatment of the disease.

#### 2.5.2. Ankylosing Spondylitis

The class of immune-mediated rheumatic diseases includes several and heterogenous disorders in which the systemic inflammation constitutes a common hallmark. Similarly, ankylosing spondylitis (AS), a type of arthritis, is characterized by chronic inflammation that primarily affects the spine and sacroiliac joints [115]. This inflammation gradually spreads and results in a progressive bony fusion called ankylosis, causing pain and stiffness at the spine and other peripheral joints [116,117]. Although the exact etiology of AS is thus far unknown, this condition is likely caused by a combination of environmental and genetic factors [118,119]. Intriguing studies have shown the involvement of miRNAs in the pathology of AS [120], and the expression profile of the major components of miRNA biosynthesis machinery have been examined. The results show a significant reduction in Dicer expression level in patients with AS (Figure 6) and a downregulation of *DGCR8* mRNA, but no effect regarding *Drosha* mRNA. Nevertheless, all three components seem to have no casual correlation with disease progression [115]. Dicer may have an impact on AS occurrence via either the miRNA functions or other underlying mechanisms. Several miRNAs have been found to be involved in processes from the regulation of immune cells and inflammatory response to ossification and osteoblasts differentiation, thus the deregulation of their expression due to the lack of Dicer could lead to an immune-associated disorder. In accordance, Treg cell function has been found to be safeguarded by a Dicer-dependent miRNA pathway [121] and miRNA aid to the maintenance of Treg cell functional program. This observation was corroborated by the fact that Dicer-deficient Treg cells lacked repression activity in vivo, and that Dicer-lacking mice rapidly progressed a lethal systemic autoimmune disorder [122]. Additional functions of Dicer enzyme that have recently been discovered, such as the involvement of the protein in autophagy [31,32,123], could also be responsible for the disease development or/and progression, as autophagy has been linked to several rheumatic diseases, including AS [124].

#### 2.5.3. Rheumatoid Arthritis

Dicer enzyme is also involved in other immune-mediated rheumatic diseases such as rheumatoid arthritis (RA), an autoimmune disorder affecting many people worldwide. In particular, RA is a systemic inflammatory disease characterized by progressive joint inflammation, leading to systemic complications and disability, as well as high morbidity and mortality rates [125,126]. This condition occurs as a result of the interaction between genetic and environmental cues, as genetic variants and epigenetic alterations increase the risk of developing the disease [127,128,129]. *Dicer* mRNA expression has been found to be significantly reduced in fibroblast-like synoviocytes (FLSs) of patients but not in isolated peripheral blood mononuclear cells (PBMCs) [130], a fact that indicates that differential Dicer expression in patients is located in specific cells (Figure 6). The latter observations are supported by a study on another group of RA patients that also revealed no statistically significant Dicer expression difference in PBMCs when compared to that of healthy individuals [131]. Thus, Dicer might contribute to the pathogenesis by controlling the differential expression of several miRNAs in RA patients, leading to the regulation of immune response, cell cycle and apoptosis [132,133]. Experimental data from mouse models with limited expression of Dicer did not demonstrate an inflammatory phenotype at first, but a significant increase of joint inflammation was observed after the injection of K/BxN sera, which can induce arthritis in normal mice [130]. Thus, it seems that the absence of Dicer may not be responsible for the occurrence of the disease but for its progression, as the patients may fail to recover from a severe inflammation. Furthermore, the increased expression level of IL-6, a pro-inflammatory cytokine, was observed in human synoviocytes upon Dicer downregulation, as well as resistance of these cells to apoptotic phenomena [130], contributing to the development of the pathological condition.

#### 2.5.4. Multiple Sclerosis

Multiple sclerosis (MS) is a chronic demyelinating disease of the central nervous system that can lead to long-term disability [134,135]. Given that Dicer expression is required for the normal development of oligodendrocytes and Schwann cells [136,137], and it is a subject of acute regulation in vitro and developmental regulation in vivo [138,139,140,141,142], many studies have focused on its implications in MS (Figure 6). Dicer also has an involvement in the development of MS, due to its crucial role in immune regulation [143]. Dicer protein levels have been found to be significantly lower in leucocytes isolated from MS patients, while there has been an inverse correlation between the protein levels and disability status of these patients [142,144]. The observed differential Dicer protein expression is not always accompanied by a corresponding reduction in mRNA levels [142], probably suggesting a post-transcriptional regulation of Dicer in these patients. Noticeably, Dicer expression is selectively decreased in the B-cells (a cell type with a fundamental role in MS) of patients with this condition, as there is no significant alteration of protein levels in other cell types [144]. Loss of Dicer leads to alterations in antibody repertoire and augmentation of self-reactive auto-antibodies in mice [145,146], which can trigger MS. In addition, Dicer may act via the molecule CD80, which is increased in a lack of Dicer [144]. CD80 is expressed by B-cells and other antigen-presenting cells and promotes the activation of T cell immune response, which is crucial in MS pathogenesis. Finally, Dicer is likely associated with the response to therapy and clinical course of MS. The treatment of patients with IFNβ1a has shown that the drug can reverse the low protein levels in some of them—probably targeting protein’s post-transcriptional modifications—which is directly related to good clinical response [142]. Hence, Dicer seems to be strongly correlated to both disease development and clinical response; consequently, it could be exploited as either a diagnostic and prognostic biomarker and/or therapeutic target for the management of MS.

#### 2.5.5. Autoimmune Thyroid Diseases

Autoimmune thyroid diseases (AITDs) are organ-specific disorders that affect approximately 5% of the general population. Such disorders include Graves and Hashimoto diseases that are characterized by the development of hyperthyroidism and hypothyroidism, respectively [147,148]. The etiology of these conditions seems to be multifactorial, as both environmental exposure and specific genes have a strong impact on it. Polymorphisms in genes involved in miRNA biogenesis, including *Dicer*, increase risk of AITDs (Figure 6). In particular, two polymorphisms in the *Dicer* gene, Dicer SNP1 (rs3742330 A/G) and Dicer SNP2 (rs1057035 C/T), which are located in the 3′ UTR, have been examined in the context of AITDs. Significantly lower frequency of the TT genotype and T allele of Dicer SNP2—but not Dicer SNP1—were reported in Graves disease patients compared to healthy individuals [149]. Dicer SNP2 most likely affects the binding of the evolutionarily conserved hsa-miR-574-3p, resulting in a decreased expression of Dicer in the absence of the T allele [150]. The correlation of lower Dicer expression levels to AITDs has also been confirmed experimentally in animal models, as Dicer-deficient mice developed uncontrollable autoimmune diseases [122]. Therefore, correcting the expression levels of Dicer might provide a crucial contributor to AITD treatment.

### 2.6. Dicer in Cardiovascular Diseases

Various studies have revealed the involvement of Dicer in a wide range of cardiovascular diseases, as it is vital for endothelial cell function [151,152] and angiogenesis [153,154]. Its absence results in the manifestation of several features of heart failure, often leading to a significant decline in cardiac function. Previous experiments in mice have shown an increased mortality rate and a decreased spontaneous activity of Dicer-deficient mice, whereas Dicer depletion has provoked geometric and functional deteriorations of the heart. Typical examples of these aberrations are increased cardiac size, thrombin conduction defects, alterations in myocardial structure and the decreased integrity of cardiomyocytes, all features that significantly reduce cardiac function [155,156]. Accordingly, a large reduction of Dicer protein levels was recorded in failing human heart tissue, indicating a role of Dicer in heart failure in human patients [156].

Importantly, Dicer enzyme affects the progression and severity of atherosclerosis and ischemia in mouse models. Downregulation of Dicer in lesional microphages induces the severity of atherosclerosis in aortas. The condition was characterized by increased inflammation, apoptosis and the presence of foam cells and large necrotic core areas, suggesting an atheroprotective role of Dicer. Furthermore, Dicer affects mitochondrial energy metabolism in macrophages, promoting the mitochondrial respiration and oxidative metabolism of fatty acids, thereby providing metabolic adaptation and preventing the progression of atherosclerosis [157]. Nevertheless, Dicer downregulation is beneficial in the case of renal ischemic injury, as Dicer-deficient mice were more resistant to ischemic acute kidney injury (AKI). In these animals, there was less tissue damage, fewer injured tubules and apoptotic cells and, generally, less severe injury [158]. It is noteworthy that SNPs in miRNAs and miRNA biogenesis pathway genes, such as *Dicer*, can increase coronary artery disease (CAD) risk. Dicer rs1057035 T > C polymorphism, located within 3′ UTR of *Dicer*, affects the probability of developing the disease, as Dicer rs1057035 CC genotype confers a 50% reduced CAD risk compared to individuals with TC or TT genotype [159]. The contribution of Dicer can be direct or indirect via miRNA regulation, whose expression has been found to be altered in these conditions [157,159]. Concluding, Dicer appears to have a crucial role in proper heart function, as its aberrant expression can lead to severe heart failure and the development of various cardiovascular diseases.

### 2.7. Dicer in Female and Male Fertility

The regulation of fertility is under strict gene control and includes the establishment of a suitable environment and the adaptation to changes in hormones and other external cues in order to produce fertilizable gametes and achieve fertilization and fetal development [160,161,162]. Several studies have demonstrated the importance of Dicer in both female and male germ cell maturation and fertility [163,164].

Loss of Dicer causes defective germ cell differentiation and reproductive system abnormalities in both sexes [165,166,167,168]. In mice, downregulation of Dicer in oocytes—a cell type containing 10–15 fold higher levels of *Dicer* mRNA than other cells and tissues—has resulted in the formation of multiple spindles and chromatin condensation defects [163,169], whereas Dicer-deficient female mice proved to be infertile [170]. Loss of Dicer also has several effects in oviduct and uteri phenotype and function in mouse models. In the absence of the protein, a shortened oviduct tubule length, loss of oviductal coils, loss of the smooth muscle layer and disorganization of the epithelium, as well as a smaller uterus in length, diameter and weight have been observed, among other responses [171,172,173,174]. In these cases, embryos were developmentally delayed and unable to enter the uterus, whereas the uterus itself was unable to sustain pregnancy following embryo transfer [171,172,174]. In *Caenorhabditis elegans*, Dicer function must be inhibited for successful oogenesis, and Dicer function must be reinitiated for embryogenesis to occur normally. An interesting explanation for this phenomenon is that specific maternal RNAs are protected in germ cells during development and degraded upon requirements via the production of certain classes of endo-siRNAs in a Dicer-dependent manner [175].

Dicer deficiency also affects the male reproductive system, as its loss leads to a severe disruption of spermatogenesis, with a typical failure of its occurrence during haploid differentiation. Chromatin condensation, abnormal head shape and disrupted organization of tail accessory structures in late spermatids, as well as defects in acrosomes and a reduction in the size of the testes by about half have been reported [165]. All the aforementioned lead to the production of abnormal gametes and failed germ cell maturation, resulting in both female and male infertility. Nevertheless, by gaining a better understanding of the role and function of Dicer in these biological pathways, this molecule could emerge as a novel tool for exploitation in the handling of infertility and other reproductive diseases.

### 2.8. Dicer-Dependent Antiviral Defense Mechanism

Viral infection in diverse eukaryotic hosts triggers the production of virus-derived small RNAs (vsiRNAs) capable of silencing mRNAs via the RNAi mechanism [176]. Such vsiRNAs have been found in fungi, plants, nematodes, insects and invertebrate animals [177,178]. The first experimental evidence of RNA-based antiviral immunity in animals was provided by the accumulation of Flock House virus (FHV) siRNAs in infected *D. melanogaster* cells [176,179]. vsiRNAs are produced specifically by the Dicer family of class-3 RNase III enzymes [180]. In particular, following transcription from the virus genome, the viral pre-siRNAs are exported from the nucleus into the cytosol, where Dicer further processes them into virus-derived small RNAs of approximately 21–23 nucleotides in length. These are assembled into effector complexes to guide the suppression of virus genes necessary for virus survival [34,178,181]. Thus, the Dicer-dependent vsiRNAs determine the specificity of RNA-based antiviral immunity and, in some cases, these molecules are the dominant species of the small RNA population in an infected cell [176]. Mammalian viruses that produce vsiRNAs are the Nodamura virus (NoV), encephalomyocarditis virus (EMCV) [180,182], hepatitis C virus (HCV) and poliovirus [176], while vsiRNAs were also detected in human 293 cells as a response to either influenza A virus (IAV) or human enterovirus 71 (HEV71) [183,184]. In case of human immunodeficiency virus 1 (HIV-1), Dicer expression and function were found suppressed in macrophages [185]. Moreover, downregulation of Dicer promoted adenovirus (Ad) replication in infected cells [34], indicating that Dicer acts as a strong antiviral with a crucial role in the RNA-based antiviral immunity.

## 3. Conclusions

In conclusion, Dicer is an essential enzyme for the maintenance of physiology due to its pivotal role in several cellular processes such as regulation of gene expression, DNA damage response, cell growth and differentiation. Loss or aberrant expression of Dicer contributes to the development of severe human diseases (Table 1), with Dicer1 syndrome being a typical example. Among other Dicer-associated disorders, some autoimmune, neurological, reproductive and cardiovascular diseases have been identified, indicating the universal action of this protein in the human body. The most obvious cascade of events through which Dicer is responsible for the development of such disorders is probably in the deregulation of miRNA expression. However, other functions of Dicer should not be overlooked, as the involvement of the protein in processes such as apoptosis, transcriptional regulation and autophagy could result in the development and progression of the disease. Notwithstanding the wealth of information already generated within this field, several open avenues for future investigation remain. The identification of new regulatory domains (including binding partners), SNPs and mutations, as well as characterizing the tissue- and cell-specific Dicer functions will likely typify future studies regarding Dicer’s biological actions. An in-depth investigation of the molecular mechanisms underlying the aforementioned pathologies could highlight promising diagnostic/prognostic tools and therapeutic targets and lead to the development of more effective, novel, Dicer-based diagnostic and therapeutic strategies, resulting in several clinical benefits and better quality of life for patients.

## Figures and Tables

**Figure 1 ijms-21-07223-f001:**
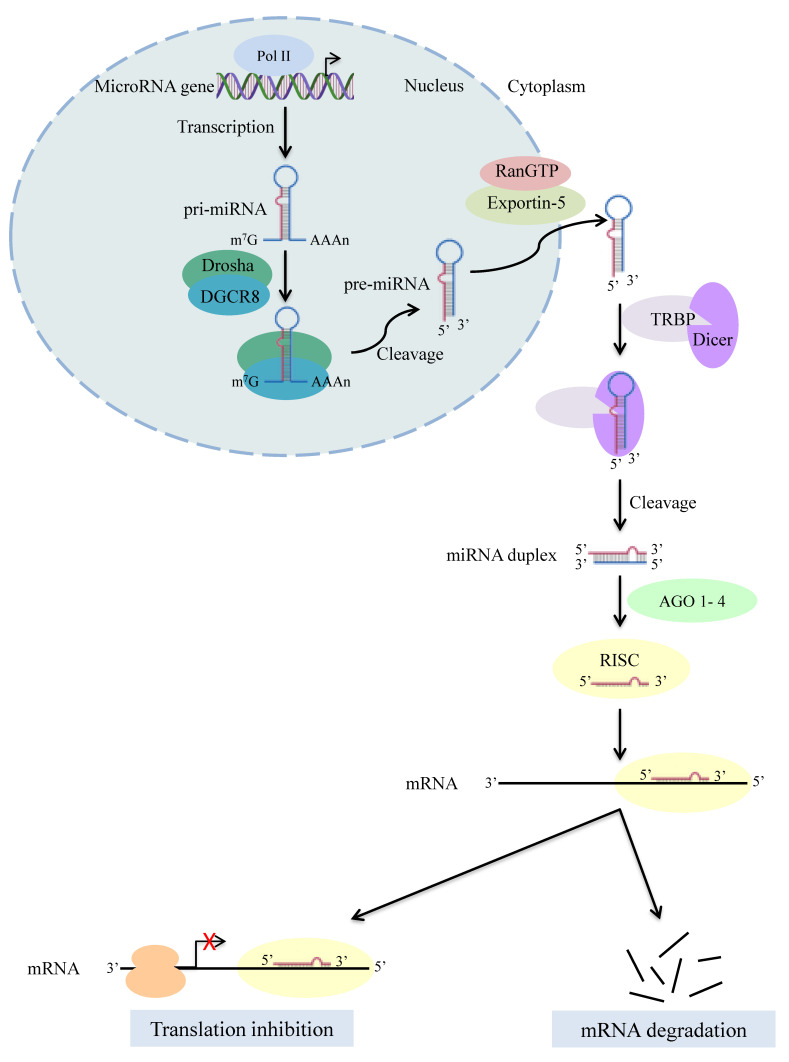
The canonical pathway of microRNA (miRNA) biogenesis. After the transcription of a miRNA gene by RNA polymerase II, the produced primary miRNA (pri-miRNA) is cleaved by the microprocessor complex Drosha-DGCR8, generating the precursor miRNA (pre-miRNA). Following the export of pre-miRNA from the nucleus by Exportin-5-RanGTP, the Dicer ribonuclease, in complex with TRBP (TAR-binding protein), cleaves the pre-miRNA hairpins to generate the mature miRNA, a small non-coding RNA (nc-RNA) of ~22 nucleotides in length. The functional strand of the mature miRNA is loaded together with Argonaute proteins (AGOs) onto the RNA-induced silencing complex (RISC) and it can then direct post-transcriptional repression via mRNA complementarity. Downregulation of gene expression can occur through translational repression with or without mRNA cleavage, depending on whether the miRNA has full or partial complementarity to the target mRNA, respectively.

**Figure 2 ijms-21-07223-f002:**
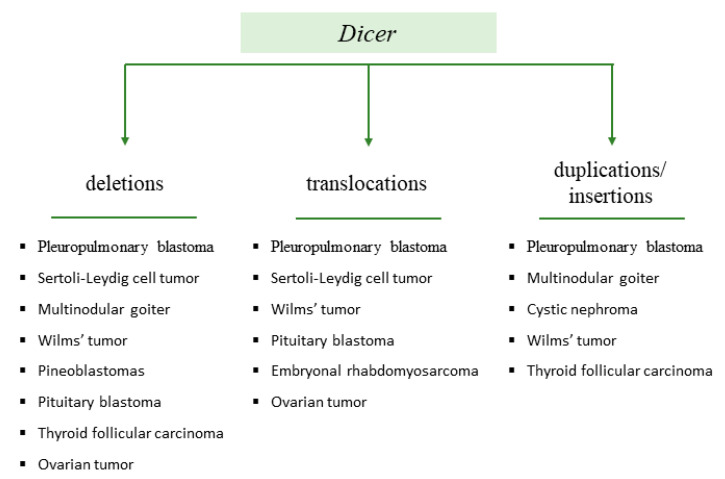
Pathogenic germline mutations in the Dicer gene lead to Dicer1 syndrome-related malignancies. Diagrammatic representation of confirmed alterations found in the *Dicer* gene (see also Figure 3B) linked to Dicer1 syndrome and predisposing the development of various cancer types [37].

**Figure 3 ijms-21-07223-f003:**
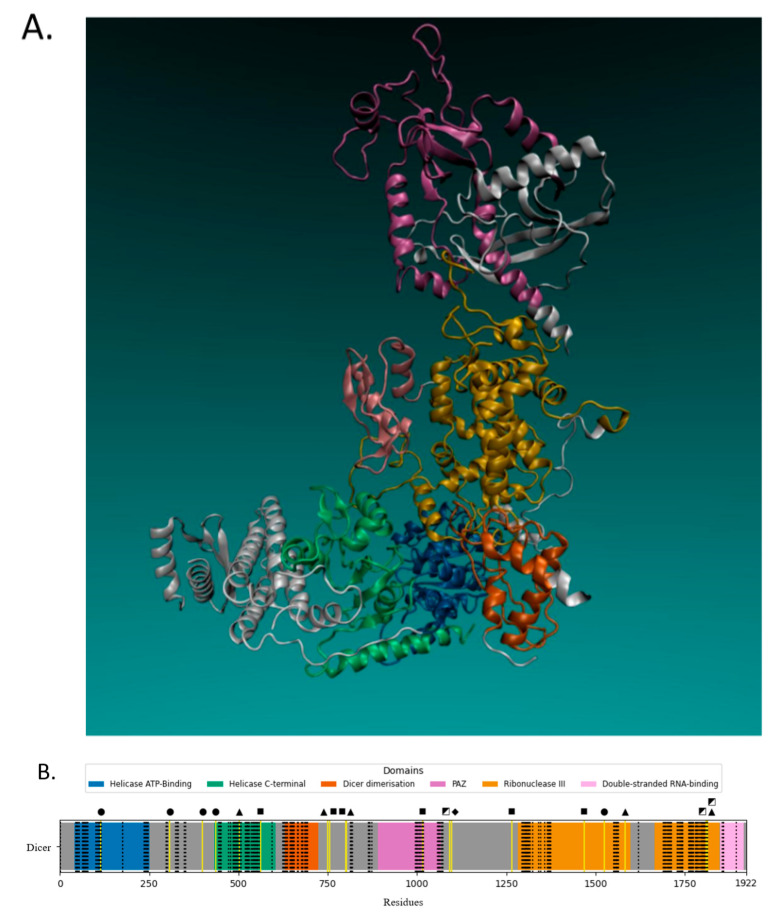
The majority of Dicer1 syndrome-related germline mutations of the *Dicer* gene are located within regions that encode the main protein’s domains. (**A**) The tertiary structure of Dicer with the protein’s main domains being highlighted (blue: helicase ATP-binding; green: helicase C-terminal; orange: Dicer dimerization; purple: PAZ; yellow: ribonuclease IIIa/b; pink: double-stranded RNA-binding). The structural data of the RCSB (Research Collaboratory for Structural Bioinformatics) PDB (Protein Data Bank) (rcsb.org) entry having the ID 5ZAK in New Cartoon representation, as displayed by VMD software. The Schrödinger Maestro Suite program (Schrödinger Release 2020-2) was applied to process the data; existing water molecules and ligands were removed and missing loops and side chains were filled (Prime). The protonation states were calculated, on physiological pH 7.4 (PROPKA), and the free energy of the resulting structures was minimized with the OPLS3 force field. (**B**) Germline mutations (yellow vertical lines) of the *Dicer* gene related to Dicer1 syndrome are presented at specific sites of the Dicer protein (■ = deletion; ◩ = deletion/insertion; ● = duplication; ◆ = insertion; ▲ = transversion). Domain information was retrieved from the EMBL-EBI Interpro database. Plots were generated with Matplotlib and Seaborn Python packages. The binding sites (black dashed lines) are depicted with vertical dotted lines and were calculated with Schrödinger SiteMap on default settings.

**Figure 4 ijms-21-07223-f004:**
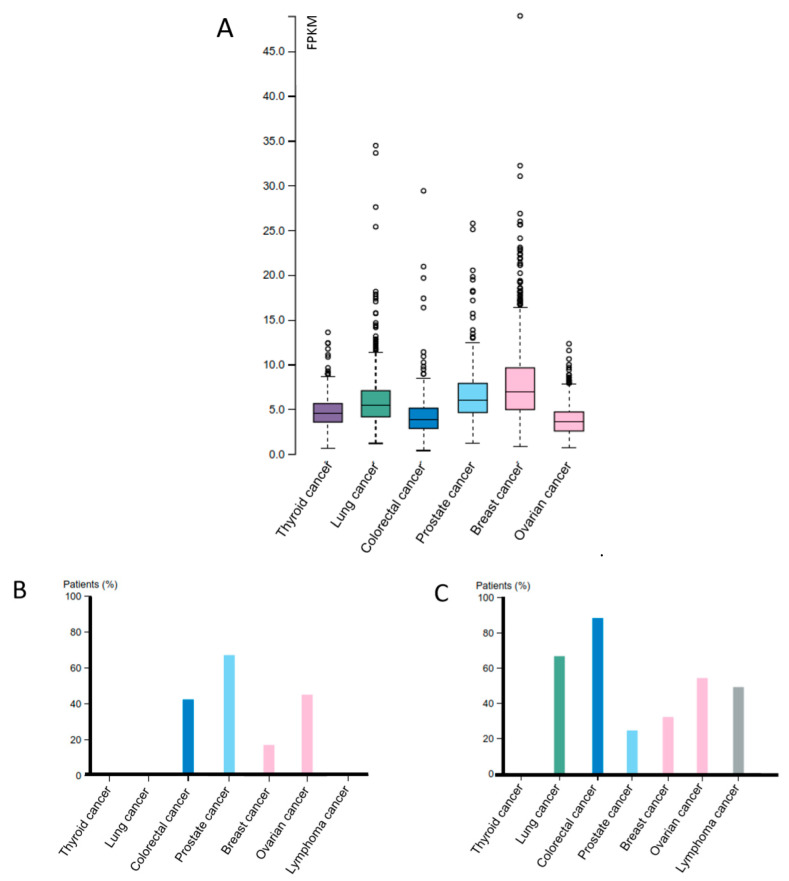
The Cancer Genome Atlas (TCGA)-derived expression profile of the *Dicer* gene in representative human cancers. Dicer expression landscaping is presented in human cancer types based on the TCGA platform (modified from TCGA). (**A**) RNA-seq data for Dicer RNA expression in thyroid, lung, colorectal, prostate, breast and ovarian cancer. The value 0.0 on the vertical axis represents zero expression of *Dicer*, while the value 45.0 maximum expression. (**B**,**C**) Dicer protein expression in thyroid, lung, colorectal, prostate, breast and ovarian cancer and lymphoma. The color bars represent the percentage of patients with medium and high protein expression levels, while the absence of a bar represents low and zero protein expression levels. The expression of the protein was evaluated with (**B**) the HPA000694 antibody (Sigma-Aldrich, St. Louis, MO, USA) and (**C**) the CAB068185 antibody (Sigma-Aldrich, St. Louis, MO, USA).

**Figure 5 ijms-21-07223-f005:**
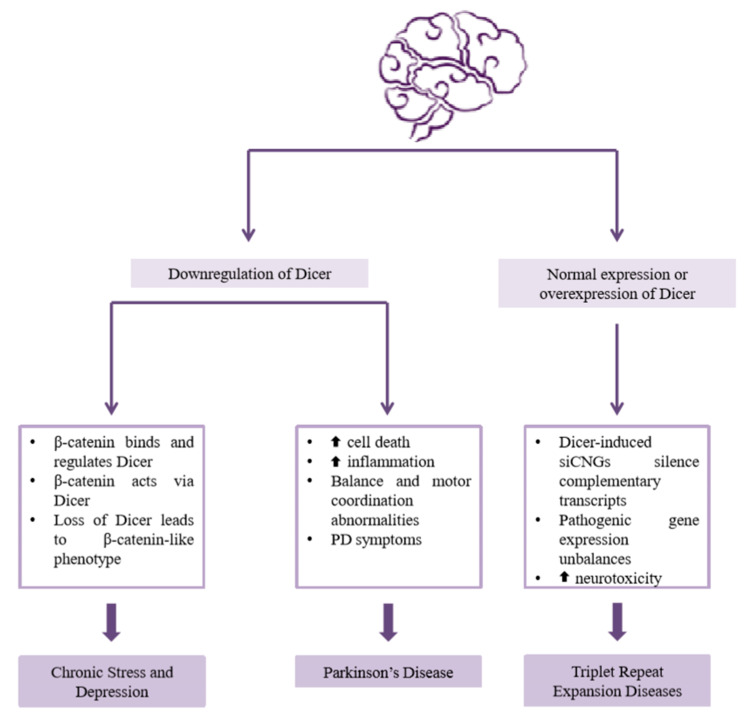
Dicer involvement in psychiatric and neurological diseases. Diagrammatic representation of the Dicer-associated neurological and neuropsychiatric disorders, as well as the typical pathways through which Dicer acts. Dicer downregulation can lead to the development of psychiatric and neurodegenerative diseases, such as depression and Parkinson’s disease, respectively, while the presence of Dicer enzyme may impair health of triplet repeat expansion disease (TRED) patients.

**Figure 6 ijms-21-07223-f006:**
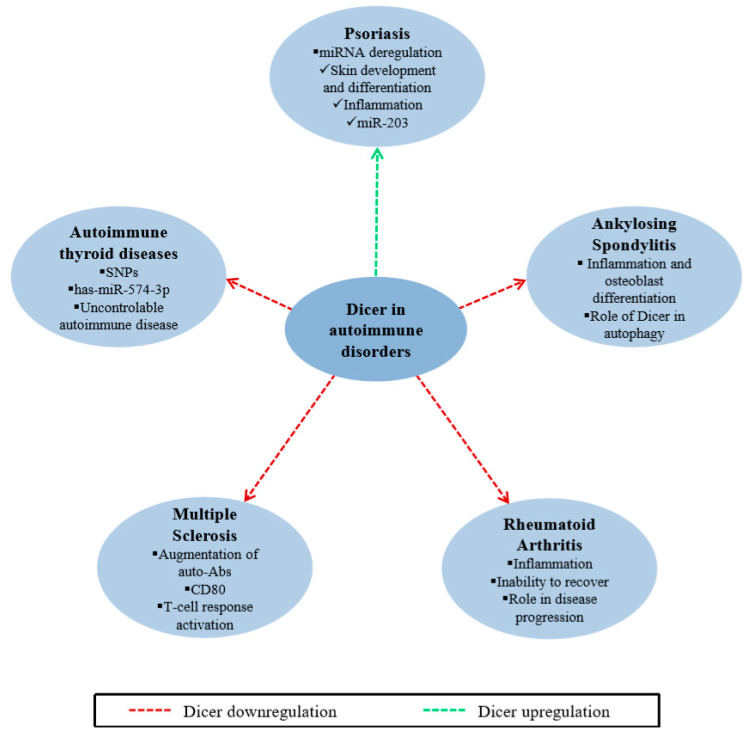
Aberrant expression of Dicer leads to the development of autoimmune disorders. Graphical overview of the main Dicer-associated autoimmune diseases. Both loss and upregulation of Dicer protein expression can cause severe autoimmune disorders, such as psoriasis, ankylosing spondylitis, rheumatoid arthritis, multiple sclerosis and autoimmune thyroid diseases.

**Table 1 ijms-21-07223-t001:** The deregulated Dicer expression in associated human pathologies.

Disease		Association
Dicer1 syndrome		germline mutations
Cancer		downregulation/upregulation
Geographic atrophy (GA)		downregulation
Psychiatric and Neurological Diseases	Chronic stress and depression	downregulation
	Parkinson’s disease (PD)	downregulation
	Triplet repeat expansion diseases (TREDs)	normal expression/overexpression
Autoimmune Disorders	Psoriasis	upregulation
	Ankylosing spondylitis (AS)	downregulation
	Rheumatoid arthritis (RA)	downregulation
	Multiple sclerosis (MS)	downregulation
	Autoimmune thyroid diseases (AITDs)	downregulation
Cardiovascular diseases		downregulation/polymorphism
Infertility		downregulation
Viral infections		vsiRNAs generation

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
