# Peer review of "Dicing the Disease with Dicer: The Implications of Dicer Ribonuclease in Human Pathologies"

_ijms, 2020, doi:10.3390/ijms21197223_

Round 1

Reviewer 1 Report

In the present manuscript, entitled- Dicing the Disease with Dicer: The Implications of Dicer Ribonuclease in Human Pathologies- by Eleni I. Theotoki and colleagues reviewed the current literature the role of Dicer in various human diseases. The authors presented decent information on the precise role of Dicer. I feel a Table related to Dicer associated diseases is missing. Also, microRNA independent functions of Dicer are missing. The figures are not pretty.

Reviewer 2 Report

The authors have put forth considerable effort to collate this paper discussing for the most part, the non-neoplastic implications of Dicer1 which are less well covered in the literature. It should be noted however that most of the non-neoplastic conditions discussed are theoretical in nature as there does not seem to be human evidence for example, of higher depression or chronic stress levels in individuals with DICER1 germline pathogenic variation. Thus these areas are interesting for hypothesis generation but caution should be taken that a lay reader may misunderstand and believe that DICER1 germline pathogenic variation predisposes to these conditions.

Reviewer 3 Report

In this manuscript, Theotoki et al. review the human diseases that may be caused by an unbalance on Dicer expression and/or function. Given the central role of Dicer in microRNA processing, this review is timely and informative. However, the text, with its Dicer-centric focus, often omits the role of microRNAs as the ultimate effectors of the microRNA processing pathway, which may help to explain the different or contradictory symptoms of Dicer dysregulation.

Overall, the manuscript merits publication in the International Journal of Molecular Sciences if the authors address the following points:

Lines 14-15.

The summary that the authors provide about the role of Dicer makes it seem like Dicer is an RNA-biding protein that binds directly to mRNA targets to regulate their expression. The authors should provide more detail about the central role of Dicer as the enzyme that processes microRNAs.

Line 53.

The authors should specify that Dicer specifically cleaves twice the hairpin at the stem-loop boundary, generating a 2 nucleotide overhang at the 3’-end, characteristic of RNase III nucleases.

Lines 55-56

miRISC complex is formed by an Argonaute family protein and a mature microRNA. The complex that includes Dicer and Ago is the miRNA loading complex. Once Ago is loaded with the mature miRNA, Dicer dissociates and the Ago2-miRNA complex induces the silencing. The author should rectify their statement accordingly.

Lines 72-81

The authors should emphasize that microRNAs induce translation repression, deadenylation, and mRNA decay after Dicer processing and when they are associated with Ago. Without this emphasis, it may mislead the readers to think that the central role of Dicer is binding directly to mRNAs.

Lines 82-84

Dicer is not involved in the processing of tRNAs or snoRNAs. Dicer is involved in generating mature miRNAs using tRNAS and snoRNAs as precursors. Please clarify in the text.

Line 151

Please reword “rendering Dicer mRNA stability” to “increasing Dicer mRNA stability”

Lines 161-162

The lack of correlation between protein and RNA levels with the phenotype may indicate that the phenotype is indeed caused by the loss of tissue-specific miRNAs due to lack of Dicer activity. Therefore, some tissues may be more susceptible to the loss of Dicer than others. Please comment.

Lines 182-183

Again, please comment on the fact that Dicer levels have a direct influence on miRNA levels, and that gain or loss of specific miRNAs may be driving the phenotype.

Line 198

It would be a good opportunity to comment on the very well documented cases where miRNAs act as oncogenes or protect from cancer, to highlight that Dicer levels will influence miRNA and tumor outcome.

Line 270 and 337

Here the authors acknowledge that the phenotype may be caused by the altered miRNA production, caused in turn after Dicer demise. This type of acknowledgment of the central role of miRNAs in driving the phenotypes should also be included in other parts of the text.
